# The Influence of the Flexibility of a Polymeric Adhesive Layer on the Mechanical Response of a Composite Reinforced Concrete Slab and a Reinforced Concrete Beam Girder

**DOI:** 10.3390/polym16030444

**Published:** 2024-02-05

**Authors:** Paweł Szeptyński, Jan Grzegorz Pochopień, Dorota Jasińska, Arkadiusz Kwiecień

**Affiliations:** Faculty of Civil Engineering, Cracow University of Technology, 31-155 Cracow, Poland; jan.pochopien@doktorant.pk.edu.pl (J.G.P.); dorota.jasinska@pk.edu.pl (D.J.); arkadiusz.kwiecien@pk.edu.pl (A.K.)

**Keywords:** polyurethane, flexible joints, shear stress, large deformations, analytical modeling, finite element analysis

## Abstract

This study addresses the challenges of modeling flexible connections in composite structures employing a polymeric adhesive layer. These types of connections provide a more uniform stress distribution compared to conventional rigid connectors. However, they lack standardized design rules and still require much research to sufficiently comprehend their properties. The novelty of this research lies in proposing an analytical solution to address these issues. Its aim is to investigate the influence of the stiffness of the polymer adhesive on the girder’s deflection and on the maximum stresses in both the adhesive and concrete. The analyzed composite structure consists of a reinforced concrete (RC) slab and an RC beam connected with a layer of flexible polyurethane (FPU) adhesive. Analytical and numerical approaches for the description of the mechanical response of a composite bridge girder are presented. Another objective is to validate the analytical design formulas using 3D nonlinear numerical analysis, both in the case of uncracked and cracked concrete. Seven types of FPUs are tested in the uniaxial tension test, each examined at five strain rates. The obtained data is used to predict the mechanical response of the considered girder using finite element analysis (FEA) as well as with a simplified one-dimensional composite beam theory. Fair agreement is found between the FEA results and theoretical predictions. A comparison of the results obtained for these two models is performed, and the similarities and discrepancies are highlighted and discussed.

## 1. Introduction

Polyurethanes (PUs) have been popular in use since the 1940s [1], in a very broad range of commercial and industrial fields, especially in those chemical, textile [2], automotive [3], and in civil engineering [4]. There is a large group of polymers containing urethane links in their structure. There are also plenty of possible ways to obtain urethanes [5]. The basic repetitive unit in PUs is the urethane group. Segmented polyurethanes are composed of a soft segment and a hard segment, consisting of the most common substrates like different isocyanates, polyols, other additives [6], and chain extenders or cross-linkers [4,7]. Each of the substrates also includes other chemical groups which, combined with individual synthesis paths, lead to a wide range of mechanical, chemical, and physical properties [8]. PUs can easily be designed by changing the types and quantities of isocyanate, polyol, surfactants, catalysts, fillers, and matrices during the manufacturing process to meet the desired performances [9,10]; therefore, PU products come in a wide variety, including elastomers, sheets, adhesives, coatings, and foams. Thanks to the polyurethanes’ versatility, they are applicable in various fields. In comparison to epoxies that are vulnerable to elevated temperatures, different PUs show that they usually do not disintegrate below 200 °C [4] and are resistant to ageing factors present in civil-engineering uses [11]. The applications of epoxy-based and polyurethane-based FRP composites in the strengthening of shear-deficient RC beams were compared in [12]. The influence of different corrosive factors, tested on flexible polyurethanes, resulted in failure concerning metal reinforcement, rather than a polyurethane matrix [11]. PUs can be used in many applications due to their chemical adaptability [13,14,15]. Following the research mentioned above, polyurethane flexible joints (PUFJs) considered to be adhesive joints in civil engineering are assumed to be durable and stable under exploitation and elevated temperatures, especially when applied in bridge-type composite structures [16], where short-term and long-term loads activate shear stress in a grid–deck connection. Bridge-type composite structures are typically constructed from steel and concrete elements, with steel connectors of different types [17]. An alternative approach is to use a continuous connection in the form of an adhesive layer.

Polymeric adhesives have been found to be an efficient solution in the construction, repair and strengthening of concrete structures. The use of the near surface mounted (NSM) composite films attached to the concrete with the use of adhesive is gaining increasing interest [18]. The adhesive properties of polymers make them an efficient solution in the repair of damaged concrete [19] or the connection of old and freshly overlaid concrete [20]. The application of adhesives in the strengthening of pre-stressed concrete bridge deck units is described in [21]. An investigation into the use of polyurethane matrix carbon fiber composites for strengthening reinforced concrete civil infrastructure is presented in [22]. The addition of polyurethane resins and aramid fibers to concrete mixes significantly improves the mechanical properties of High-Strength Concrete (HSC), as shown in [23]. In [24], a new material, called polyurethane–cement (PUC—a mixture of cement, polyether, polyisocyanate, silicon oil and water), is employed to strengthen RC T-sections. Nowadays, timber–concrete composites (TCC) emerge as an efficient alternative to traditional steel–RC structures. The application of adhesives in connections in TCC structures is extensively investigated in [25,26,27,28]. The first attempts in the bonding of the concrete-to-concrete elements of composite structures can be found in [29,30]. The advantages and disadvantages of epoxy adhesives in concrete-to-concrete bonds are presented in [31], but no flexible polyurethane solution is mentioned there; thus, the novelty of the present solution is valid.

It is worth noting that the type of connectors used in a composite structure influences its mechanical response in a significant way, as depicted in Figure 1. If a rigid connection is used, namely such that there is no slip between the bottom surface of the plate and the top surface of the beam, then Bernoulli’s hypothesis of plane cross-sections may be used for such a composite section. On the other hand, if there is no interaction between the plate and the beam, then both sections respond to some extent independently. A flexible connection, which is obtained when a polymeric adhesive layer of sufficient flexibility is used, is the one corresponding with an intermediate situation between rigid connection and free sliding.

The use of a layer of flexible polyurethane adhesive as an innovative connection between reinforced concrete (RC) slab and RC T-section is proposed in [16]. In the referenced paper, a linear one-dimensional analytical beam model was used. However, in our current research, we broaden the scope to include two general situations: one where the concrete remains uncracked, and another where the concrete exhibits cracking. Additionally, the theoretical model’s predictions are validated with a nonlinear 3D numerical Finite Element Method model. 

An interface layer of FPU constructs the polyurethane flexible joint, transferring high loads and high deformations simultaneously. Thanks to its flexibility, it can reduce stress concentrations and redistribute them more evenly along the bond length [28]. The problem of the theoretical analysis of highly nonlinear PUFJ is not trivial. Analytical and numerical analyses of PUFJ behavior were carried out using various methods [32,33]. In the present article, a composite girder is analyzed, which consists of an RC slab and an RC beam connected with a layer of FPU adhesive. The influence of the mechanical properties of the adhesive on the girder’s deflection and on the stress distribution in adhesive and adherends is investigated. Seven types of FPU are tested in the uniaxial tension test, each examined at five strain rates. The obtained data are used to predict the mechanical response of the considered girder with the use of finite element analysis (FEA) as well as a simplified composite beam theory. Comparisons of the results obtained with these two models for cases of both uncracked and cracked concrete are performed and the conclusions are formulated.

## 2. Materials and Methods

### 2.1. Flexible Polyurethanes

Analyzed materials are elastic polyurethane adhesives PSTF-W, PM, PS, PST, PTS, PT, and PSTF-S. These are the component adhesives of the type P, distributed by FlexAndRobust Systems (Cracow, Poland). These materials tend to have hyperelastic characteristics, meaning that they have high strain and failure energies. The presented adhesives have been used successfully to repair concrete surfaces and strengthen both concrete beams and masonry structures. More advanced properties of the presented polyurethanes can be found in [34], where mechanical characterizations such as the storage moduli, obtained from DMA tests, are presented. More details concerning material properties of the considered polyurethanes can be found in [35].

### 2.2. The Experimental Setup and Testing Procedures

The properties of the polyurethane materials were determined by a uniaxial tensile test of dog-bone shaped specimens, according to ISO 527 [36]. Shape 1A specimens (Figure 2) were made by casting in silicone molds. The results of the uniaxial tensile test may be straightforwardly employed in the Abaqus/CAE 2022 FEA software in order to determine the coefficients of the assumed constitutive model.

The tests were carried out on a Zwick/Roell 1455 20 kN testing machine (Zwick Roell Polska, Wroclaw, Poland) in the laboratory of the Cracow University of Technology (Figure 3). Tensile tests were performed on seven polyurethane materials: type PSTF-W, PM, PS, PST, PTS, PT, and PSTF-S. The variable parameter of the research was the strain increment speed (10^−3^, 10^−2^, 10^−1^, 10^0^ and 10^1^ 1/min) for the measuring base of the long-range extensometer of 50 mm. Applied strain rates align with a quasi-static way of loading rather than dynamic or impact action. This corresponds with the fact that structure’s self-weight and quasi-static component of live load usually surpass the dynamic load component in typical bridge structures. Six samples of each material were examined at a temperature of 23 °C for every speed setting. This led to a total of 246 samples being tested.

### 2.3. The Results of the Uniaxial Tension Test

On the basis of uniaxial tensile testing of dog-bone shaped specimens, the described material properties of six polyurethanes PSTF-W, PM, PS, PST, PTS, PT and PSTF-S were determined in five ranges of strain rates. Mean stress–strain curves for the seven investigated polymers tested at strain rate 100%/min are presented in Figure 4. 

Viscoelastic materials exhibit higher stiffness at higher strain rates, which is illustrated in Figure 5, in which mean stress–strain curves corresponding with different strain rates are presented for polyurethane PM.

The obtained mean stress–strain curves were used to determine the initial tangent Young’s modulus of the investigated polyurethanes, calculated as a slope of a line tangent to the first part of each curve, neglecting the initial data points affected by deformation of the setup itself. Initial tangent longitudinal stiffness was chosen for the description of the investigated material according to the assumption that the considered magnitudes of strain were small, within just a few percent. The results are presented in Table 1.

The moduli’s variations in relation to strain rates, presented in Table 2, are irregular, increasing as strain rates increase from 0.1%/min to 1000%/min. Sensitivity of the polyurethanes to the strain rate is caused by various additives and components’ proportions influencing viscosity properties. The PU most vulnerable to strain rate is PM, whereas the most resistant is PST.

### 2.4. The Studied Case

In order to investigate the influence of mechanical properties of FPUs on the mechanical response of composite structures utilizing these adhesives, the case of a composite girder is studied. The girder consists of a reinforced-concrete (RC) slab 20 cm thick and an RC beam of rectangular cross-section 30 cm × 60 cm beneath the slab. The RC elements are bonded with a 2 cm thick layer of FPU. The considered girder is a section of a repetitive system of parallel girders in the axial spacing of 100 cm. Concrete is reinforced with rebars of diameter ϕ=12 mm and two-legged stirrups of diameter $\phi_{w}=6\;\mathrm{mm}$. The concrete cover is 3 cm thick. The reinforcement layout is presented in Figure 6 and Figure 7.

The span length of the girder (between the supports’ axes) is L=6 m. The beam is assumed to be symmetric, simply supported, made from C30/37-class concrete, and characterized by mean Young’s modulus $E_{cm}=32.0\;\mathrm{GPa}$ and Poisson’s ratio $\nu_{c}=0.2$. The mean compressive strength is $f_{cm}=38\;\mathrm{MPa}$, while mean tensile strength of the concrete, which determines its resistance against cracking, is equal to $f_{ctm}=2.9\;\mathrm{MPa}$. The characteristics of the reinforcement steel are $E_{s}=200\;\mathrm{GPa}$ and $\nu_{s}=0.3$.

## 3. Analytical Model

An analytical model of a multilayer composite beam used for the description of the deformation of the considered girder was proposed in [37]. It may be considered an intermediate approach between a Discrete Layer-wise Model [38] and an Equivalent Single Layer model [39], as separate longitudinal displacement is attributed to each bent layer in the composite, while both share common deflections. Due to assumptions about small strains, small displacements, and linear constitutive relations, the considered model belongs to the framework of linear theory of elasticity. The model assumes that a composite beam consists of bent layers (beams) and sheared adhesive layers placed in an alternating way. Bent layers are considered slender, hence they can be modelled with the use of the Bernoulli–Euler beam theory. The adhesive layers are assumed to exhibit much smaller rigidity than the neighboring bent layers, so that the mechanical state of the adhesive may be approximated as the simple shear. This assumption is not strictly correct, since adhesive layers which are most distant from the neutral axis of the beam are obviously stretched or compressed. However, in case of a large class of polymers, the Poisson’s ratio, of which approaches 0.5 distortional deformation, surpasses the volumetric response of the adhesive. For this reason, the model utilizing this simplifying assumption corresponds well with both experimental and FEA results, as was shown in [37].

### 3.1. Governing Equations

The system of governing the equations of the considered composite-beam theory is obtained by substituting linear constitutive relations and geometric relations, resulting from the assumed kinematics of the system, in the equilibrium equation. In the case of a girder consisting of two bent layers (beams) and a single sheared adhesive layer, these equations are as follows:(1)$$\left\{\begin{array}{l} \frac{d^{2}{\tilde{u}}_{1}}{dx^{2}}+\pi_{2}\left[\left({\tilde{u}}_{2}-{\tilde{u}}_{1}\right)+\pi_{1}\frac{d\tilde{w}}{dx}\right]=0\\ \frac{d^{2}{\tilde{u}}_{2}}{dx^{2}}-\pi_{3}\left[\left({\tilde{u}}_{2}-{\tilde{u}}_{1}\right)+\pi_{1}\frac{d\tilde{w}}{dx}\right]=0\\ \frac{d^{4}\tilde{w}}{dx^{4}}=\pi_{5}+\pi_{4}\left[\left(\frac{d{\tilde{u}}_{2}}{dx}-\frac{d{\tilde{u}}_{1}}{dx}\right)+\pi_{1}\frac{d^{2}\tilde{w}}{dx^{2}}\right] \end{array}\right.$$
where the unknown functions ${\tilde{u}}_{1},{\tilde{u}}_{2},\;\tilde{w}$ are the non-dimensional relative displacements: the longitudinal displacement of the top beam and the longitudinal displacement of the bottom beam and common deflection, respectively. They are calculated by dividing the regular displacement by a characteristic length of the system, e.g., the span length L. The parameters of this system of equations are the system’s similarity numbers, defined as the following:(2)$$\begin{array}{c} \pi_{1}=\left(\frac{h_{1}+h_{2}}{2L}\right),\;\;\pi_{2}=\frac{G_{a}L^{2}b}{E_{1}A_{1}t},\;\;\pi_{3}=\frac{G_{a}L^{2}b}{E_{2}A_{2}t}\\ \pi_{4}=\frac{G_{a}L^{3}b\left(h_{1}+h_{2}+2t\right)b}{2t\left(E_{1}I_{1}+E_{2}I_{2}\right)},\;\;\pi_{5}=\frac{L^{3}b\left[q+\left(h_{1}g_{1}+h_{2}g_{2}\right)+\left(tf\right)\right]}{E_{1}I_{1}+E_{2}I_{2}} \end{array}$$
where L is the characteristic length of the system, b and t are the width and thickness of the adhesive layer, respectively, and q is the integrated surface traction on the top surface of the girder divided by the width of the adhesive layer. The parameters denoted with subscript refer to the top beam for i=1 and to the bottom beam for i=2. Quantities $h_{i},\;A_{i},\;I_{i}$ stand for the beams’ heights, areas and the second moments of the areas of their cross-sections, respectively. Body forces due to the beams’ self-weight are denoted with $g_{i}$, while f stands for body forces related with the adhesive layer. $E_{i}$ are the Young’s moduli of the bent layers, and $G_{a}$ is the Kirchhoff’s modulus of the adhesive.

The obtained system of governing equations is an inhomogeneous linear system of three fourth-order ordinary differential equations (ODE) with constant coefficients. It may be transformed into a system of eight first-order ODEs.

### 3.2. Analytical Solutions for a Simply Supported Beam under a Uniformly Distributed Load

The methods of solving linear systems of first-order ODEs with constant coefficients are well known. In [32], an analytical solution to this problem was found with the use of the method of generalized eigenvectors [40]. Closed-form expressions for maximal deflections and maximal stresses both in the bent layers and the sheared layer may be derived for the case of a simply supported beam loaded with uniformly distributed load (UDL) asMaximal deflection:(3)$$w_{max}=L\pi_{5}\frac{768\pi_{1}\pi_{4}e^{\frac{\lambda }{2}}+\left(e^{\lambda }+1\right)\left[8\lambda^{2}\left[\pi_{1}\pi_{4}\left(\pi_{2}+\pi_{3}+6\right)+{\left(\pi_{2}+\pi_{3}\right)}^{2}\right]-3\lambda^{4}\left(\pi_{2}+\pi_{3}\right)-384\pi_{1}\pi_{4}\right]}{384\lambda^{6}\left(e^{\lambda }+1\right)}$$Extremal compressive stress in the top bent layer (in the middle of the span):(4)$$\sigma_{min}=E_{1}\pi_{5}\left[\frac{\pi_{1}\pi_{2}}{\lambda^{4}}\;\left[1-\frac{\lambda^{2}}{8}-\frac{2e^{\frac{\lambda }{2}}}{e^{\lambda }+1}\right]+\frac{h_{1}}{2L}\frac{8\pi_{1}\pi_{4}\left(2e^{\frac{\lambda }{2}}-e^{\lambda }-1\right)-\lambda^{2}\left(\pi_{2}+\pi_{3}\right)\left(e^{\lambda }+1\right)}{8\lambda^{4}\left(e^{\lambda }+1\right)}\right]$$Extremal tensile stress in the bottom bent layer (in the middle of the span):(5)$$\sigma_{max}=E_{2}\pi_{5}\left[\frac{\pi_{1}\pi_{3}}{\lambda^{4}}\;\left[\frac{\lambda^{2}}{8}-1+\frac{2e^{\frac{\lambda }{2}}}{e^{\lambda }+1}\right]-\frac{h_{2}}{2L}\frac{8\pi_{1}\pi_{4}\left(2e^{\frac{\lambda }{2}}-e^{\lambda }-1\right)-\lambda^{2}\left(\pi_{2}+\pi_{3}\right)\left(e^{\lambda }+1\right)}{8\lambda^{4}\left(e^{\lambda }+1\right)}\right]$$Maximum shear stress in the adhesive layer (in the support area):(6)$$\tau_{max}=\frac{GL}{t}\frac{{\;\pi }_{1}\pi_{5}\left[\lambda \left(e^{\lambda }+1\right)-2\left(e^{\lambda }-1\right)\right]}{2\lambda^{3}\left(e^{\lambda }+1\right)}$$
where $\lambda =\sqrt{\pi_{1}\pi_{4}+\pi_{2}+\pi_{3}}$. The general characters of the distributions of displacements and stresses along the beam length are depicted in Figure 8.

### 3.3. Accounting for the Cracking of Concrete in the Linear Analytical Model

The cracking of concrete is one of the key aspects of the appropriate modelling of bent RC structures. Since crack depth, determining the local decrease in a beam’s flexural stiffness, depends on the magnitude of cross-sectional forces which in turn depend on the flexural stiffness, the deformation of cracked RC structures is in fact a non-linear problem of elasticity. One of the ways of solving such problems is an iterative adjustment in flexural rigidity according to a determined distribution of cross-sectional forces and solving the sequence of linear problems, until the relative increments in the distribution of flexural stiffness, displacements, and cross-sectional forces in the current step related to the results obtained in previous steps do not exceed certain threshold values. Such an approach is especially useful in the FEA, in which the stiffness of each finite element can be updated independently. 

In the case of analytical modelling, such an approach fails, since the stiffness is updated locally; hence, it is impossible to use linear equations. However, it is common practice in determining the deflection of cracked elements analytically to assume a constant reduced flexural stiffness for a whole beam. In [41], it was shown that for single-span beams under UDL, in most typical design cases, the influence of the non-uniformity of cracking along the beam’s axis on its deflection is not significant. For this reason, the linear analytical model is also used in order to iteratively obtain a solution to the problem of the deformation of a cracked beam. Stress and strain distribution in a RC cross-section is determined according to the Bernoulli–Euler theory for composite cross-sections; in particular, the hypothesis of plane cross-section is assumed valid. The weighted characteristics of the RC cross-section are calculated by multiplying the contribution of each material by an appropriate ratio of the material’s Young’s modulus to the reference modulus. If, after a certain load step, the maximal tensile stress in concrete is found to be larger than mean tensile strength $f_{ctm}$, updated weighted geometric characteristics of cross-sections are calculated in such a way that this part of the concrete section in which stress exceeds $f_{ctm}$ is disregarded in the calculation of the surface area and the second moment of the RC cross-section. The iterations are repeated until the relative change in the geometric characteristics does not exceed a threshold value.

## 4. Finite Element Analysis

The comparative finite element analyses of the loading process of the composite girders with different FPU adhesives have been conducted with the Abaqus/CAE 2022 software package.

### 4.1. Geometry, Supports, and Loads

The geometry, support placement, and the reinforcement layout of a three-dimensional finite element model of the composite girder, presented in Figure 9, accurately reflect the original design. 

The reinforcement bars (both rebars and stirrups) are modelled as individual rods (truss elements) embedded in the concrete members’ elements. The upper and lower surfaces of the adhesive layer are tied to the concrete slab and beam, respectively. Unilateral contact conditions introduced between the beam and the rigid supports, allowing for both translation and rotation, simulate the simple support conditions. The self-weight and the UDL (applied between the axes of the supports) constitute the girder loading. 

### 4.2. The Mechanical Propereties of Materials

FPU is assumed to be hyperelastic, modeled with the Mooney–Rivlin form of strain energy potential, with coefficients calibrated separately by the uniaxial tension test data for each analyzed polyurethane at every deformation speed, as described in Section 2.3. Such a procedure is available in Abaqus. The compressibility of the FPU is accounted for by assuming the Poisson’s ratio equal to ν=0.4, according to [42,43].

The smeared crack concrete model has been chosen to represent the behavior of concrete after reaching the tensile strength. This model uses isotropic hardening in compression and the idea of a crack detection surface and oriented damaged elasticity concept to represent cracking in tension. The ratio of failure stresses in tension and compression has been assumed to be 0.0763. The tension stiffening describing the post-failure behavior for straining across cracks is defined as the stress–strain two-point curve, in which post-failure stress reaches zero for the strain across the crack equal to 0.001. Parameters describing the compressive concrete behavior are not listed as they are irrelevant, since in all analyzed cases the compressive stresses in concrete are below 8% of the compressive strength. For the same reason, the steel material of the reinforcement is assumed to be linearly elastic. The elastic and strength data for steel and concrete are assumed as in Section 2.4.

## 5. Results

Figure 10, Figure 11, Figure 12, Figure 13, Figure 14, Figure 15 and Figure 16 present a comparison between the results obtained from an analytical model (AN) and from FEA for the case of an uncracked girder loaded with UDL of magnitude $5\;\mathrm{kN}/m^{2}$. Please note that, due to the superposition principle, which is valid within the framework of linear theory of elasticity, the results obtained from the linear elastic analytical model may be simply scaled unless cracking occurs. It is not the case with regard to the FEA results, since nonlinear constitutive relations of the Mooney–Rivlin hyperelastic material were adopted for the adhesive, according to the obtained experimental results. The blue bars show the results from the analytical model, while the red bars correspond with the FEA results. Color intensity is related to the strain rate of the uniaxial tension test for which the adhesive properties of the Young’s modulus in the analytical model and the Mooney–Rivlin coefficients in FEA were investigated.

The FEA results obtained for PST, PSTF-W and PS adhesives at strain rate 0.1%/min were excluded from the below comparison, due to the fact that the mean strain–stress curves corresponding with such small strains were flawed by the deformation of the experimental setup and by insufficient sampling frequency in these cases.

In Table 3, the magnitudes of UDL for which first cracking occurs are presented for each type of adhesive (tested at strain rate 100%/min) for both the analytical model and the FEM model.

Simulations have also been carried out to validate the iterative approach applied for the analytical model in order to take into account cracking of concrete. A single case was investigated, namely a girder with a sheared layer made of PT polyurethane, (tested at strain rate 100%/min) loaded with the UDL of 20 kN/m^2^. After 15 iterations, increments of deflection, extremal normal stresses in the plate, shear stress in the adhesive and the values of the weighted geometrical characteristics of cross-sections were less than 1%. The results are presented in Table 4.

## 6. Discussion

According to the obtained results one may notice that the following is true:Uniaxial tensile tests show a dependence of the strain increment speed and the change in properties of the analyzed polyurethane materials—higher strain rates result in larger values of the initial tangent Young’s modulus. Analyzing the short-term load at a strain rate of 1000%/min, the material’s initial tangent Young’s modulus E was 118% larger than that of the long-term load at a strain rate of 0.1%/min for PM polyurethane;The analytical model underestimates the maximal deflection. The relative difference is 5–20%. The greatest discrepancies may be observed for stiff polyurethanes;The analytical model underestimates the maximal distortional strain in the adhesive layer. The relative difference is ca. 15–50%. There are two main reasons for such a large discrepancy. The first is that maximal shear stress occurs in the support zone in which the strain state becomes a complex three-dimensional state—it cannot be properly modelled by a one-dimensional beam model. The second reason is that, in the beam model, it is assumed that the adhesive layer undergoes pure shear only, so the volumetric response of the adhesive is totally neglected;The analytical model overestimates the stresses both in the adhesive and the concrete. Regarding the shear stress in the adhesive, the relative difference in most cases of flexible adhesives is ca. 5–20%; however, in the case of stiff polyurethanes (PSTF-S, PT), the difference may be as high as 30–80%. Regarding concrete, the relative difference is approximately 10–30% for maximal tensile and minimal compressive stress in the plate. It is less than 10% for both maximal tensile stress and minimal compressive stress in the beam in the case of more flexible adhesives; however, in the case of more stiff polyurethanes, the differences in compressive stresses may be as high as 25%. Some individual cases fall beyond these general estimates. Qualitatively similar findings regarding normal stress in bent layers were reported in [32,37];An overestimation of the stresses in the analytical model results in the fact that cracking occurs for smaller magnitudes of the ULD, compared to the FEA; however, both estimates are similar;The use of more stiff adhesives results in smaller deflection, smaller distortional strain in the adhesive, larger shear stress in the adhesive, and smaller normal stresses in concrete—this relation is reproduced by both the analytical model and FEA. This conclusion concerns both the change in the adhesive’s stiffness related with the type of polymer, as well as with the strain rate;In case of the investigated case of the deformation of the cracked girder, the iterative solution of the linear problem with the use of an analytical model, assuming the constant reduction of the flexural stiffness over the whole beam length, gives a fair approximation of the deflection calculated with the use of the nonlinear FEM model. The stress estimates are of much lower accuracy.

## 7. Conclusions

Some significant discrepancies between the analytical and numerical models may be observed for certain cases (e.g., shear stress in stiff adhesives). There are three main sources of these differences: (1) in the beam model, Bernoulli’s hypothesis of plane cross-sections is assumed, while the 3D FEM model is not constrained in such a way; (2) in the analytical model, the adhesive layer undergoes simple shear only—it disregards the transverse contraction and longitudinal elongation of the adhesive, while these phenomena are accounted for in the FEM model; and, (3) in the analytical model, the constitutive relations for the adhesive are assumed to be linear (Hooke’s law), while in the numerical model the nonlinear hyperelastic Mooney–Rivlin constitutive model is introduced.

Despite the fact that estimates provided by the analytical model are safe with regard to stresses, they may lead to an uneconomical design. The analytical model could be improved by taking into account the longitudinal deformation of the adhesive layer—it may be conducted in a way similar to that proposed in [44]. The problem then becomes much more computationally complex, since it requires distinguishing between the deflection of the bottom bent layer and the deflection of the top one. This increases the number of equations in the first order system of ODE by four. Finding a closed-form analytical solution for such an extended system encounters significant difficulties even with use of computer algebra software—the computational effort is comparable with that required for FEA, which makes this approach rather impractical.

The analytical and numerical models presented in this research do not include visco-elasto-plastic phenomena as well as the durability aspects mentioned in [4,11]. Polyurethane degradation caused by thermal [45,46,47,48] and other [49,50,51,52,53,54,55] factors should also be included in the above presented analysis in the future.

When it comes to the description of the deformation of girders in which the cracking of concrete occurs, the obtained results look promising with regard to the estimate of the deflection of the structure. In order to formulate more general and conclusive statements regarding the applicability of the considered analytical model in the analysis of cracked structures, a more detailed and comprehensive analysis must be performed. The problem of the deformation of a material undergoing cracking is nonlinear; therefore, it is necessary to consider multiple load levels in the analysis, as the principle of superposition is no longer valid.

The proposed analytical model provides closed-form expressions for most important quantities taken into consideration during the design process: maximal deflections and extreme stresses in the concrete elements and the adhesive joint. The discrepancies between the simplified analytical model and the detailed numerical approach indicate that these design formulas may be used as an initial estimate description of a structure’s response, since they do not require the construction of any sophisticated design model. These equations emerge as a useful design tool suitable for flexible joints which lack standardized design rules, as opposed to typical rigid connectors (e.g., EN 1994 [56]). The proposition first communicated in [16] still requires much research, primarily the experimental testing of reduced-scale and full-scale composite specimens.

## Figures and Tables

**Figure 1 polymers-16-00444-f001:**
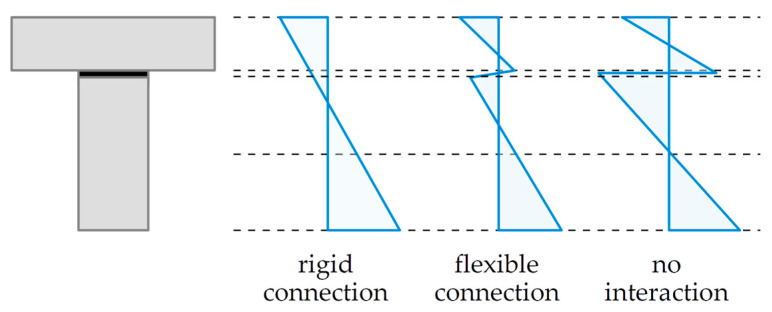
Strain distribution in composite cross-sections, depending on the type of connection.

**Figure 2 polymers-16-00444-f002:**
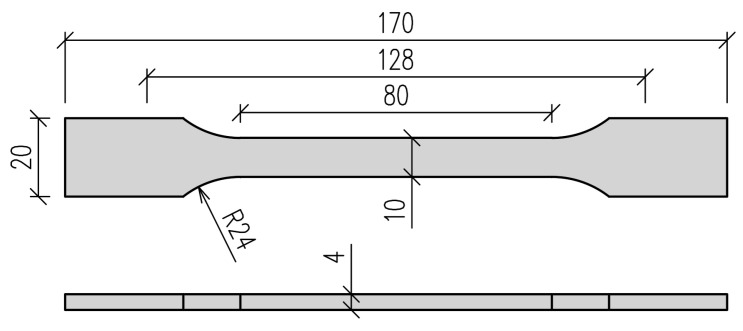
Geometry of specimen type 1A according to ISO 527 standard.

**Figure 3 polymers-16-00444-f003:**
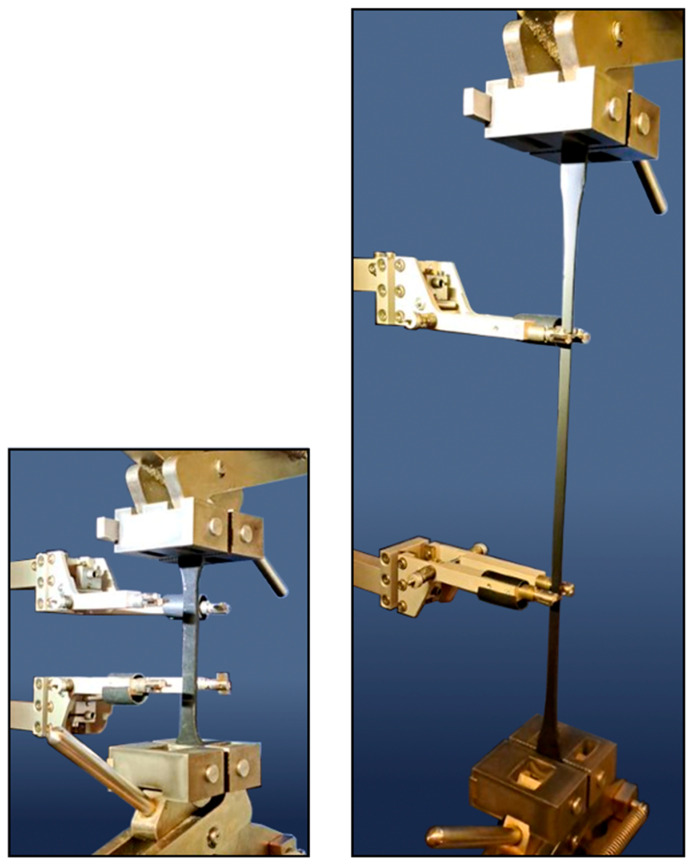
Tensile test experimental setup. (**Left**): before the specimen deformation, (**Right**): after the specimen deformation.

**Figure 4 polymers-16-00444-f004:**
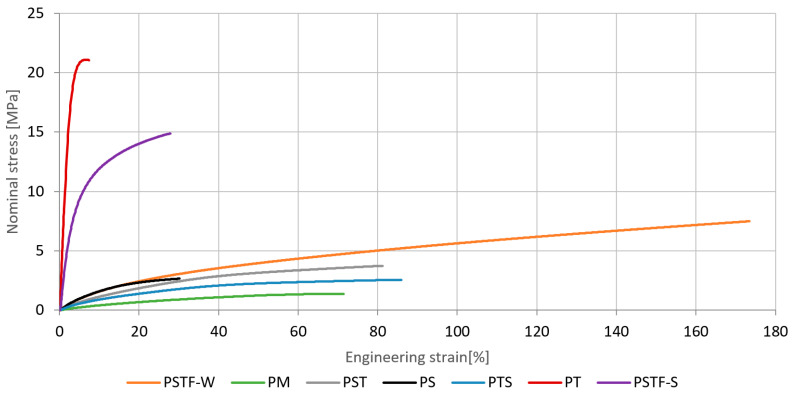
The characteristics of the polyurethanes for the strain rate 100%/min—mean values.

**Figure 5 polymers-16-00444-f005:**
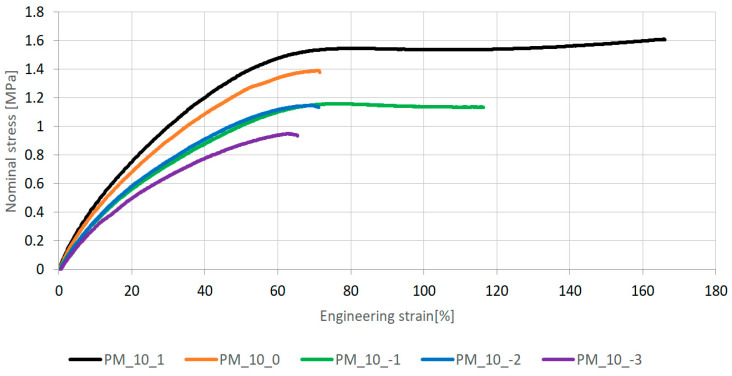
The characteristics of the polyurethane PM and its dependence on the strain rates—mean values.

**Figure 6 polymers-16-00444-f006:**
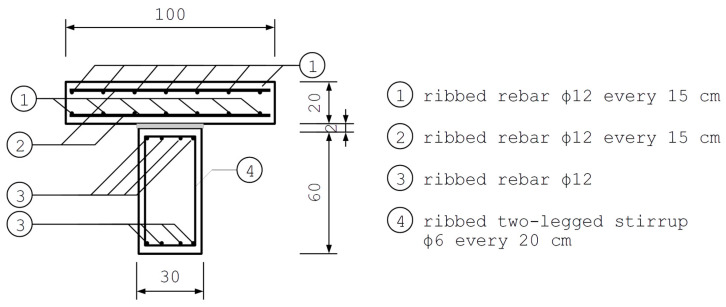
Cross-section of the composite girder.

**Figure 7 polymers-16-00444-f007:**
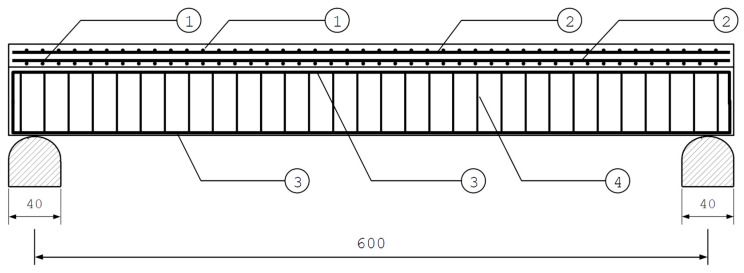
Side view of reinforcement in the composite girder (the explanation of the numbers is presented in Figure 6).

**Figure 8 polymers-16-00444-f008:**
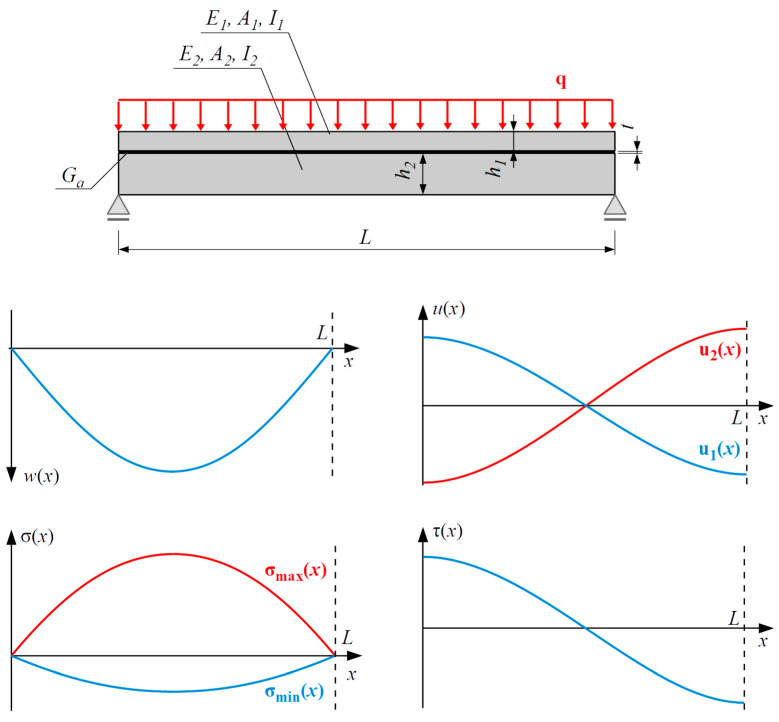
Distributions of deflection $w\left(x\right)$, horizontal displacements $u_{1}\left(x\right)$ and $u_{2}\left(x\right)$, extremal tensile and compressive stress in concrete $\sigma_{max}\left(x\right)$ and $\sigma_{min}\left(x\right)$, and shear stress in adhesive layer $\tau \left(x\right)$.

**Figure 9 polymers-16-00444-f009:**
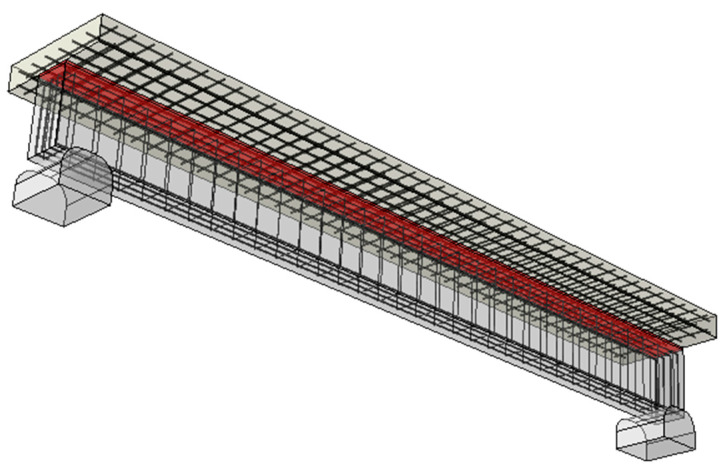
FEM model of the composite girder (the red color marks the FPU adhesive layer).

**Figure 10 polymers-16-00444-f010:**
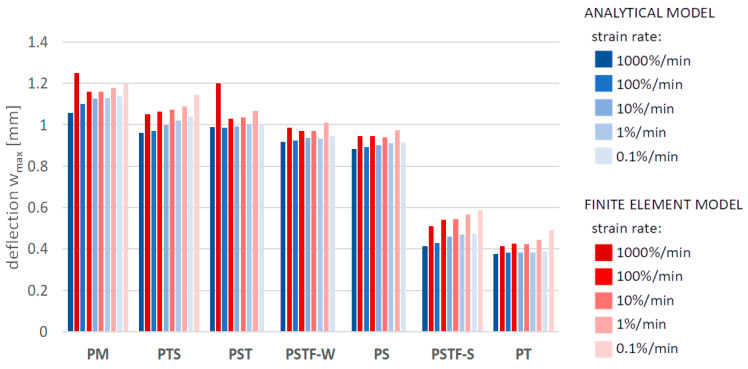
Maximum deflection in the middle of the span.

**Figure 11 polymers-16-00444-f011:**
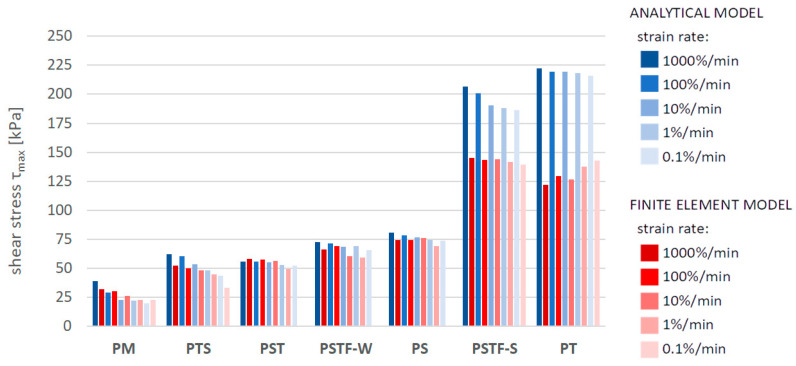
Maximal shear stress in the adhesive layer in the support zone.

**Figure 12 polymers-16-00444-f012:**
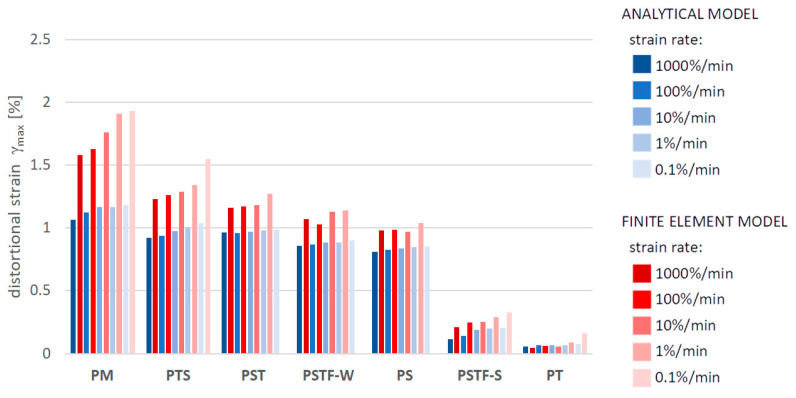
Maximal distortional strain in the adhesive layer in the support zone.

**Figure 13 polymers-16-00444-f013:**
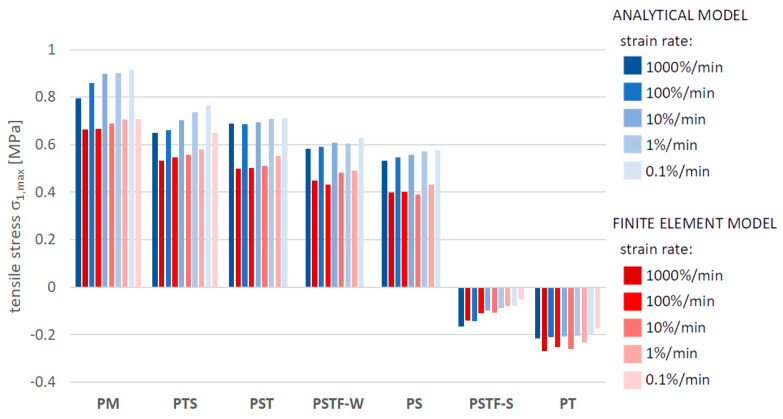
Maximal stress in the mid-span in the bottom fibers of the top plate.

**Figure 14 polymers-16-00444-f014:**
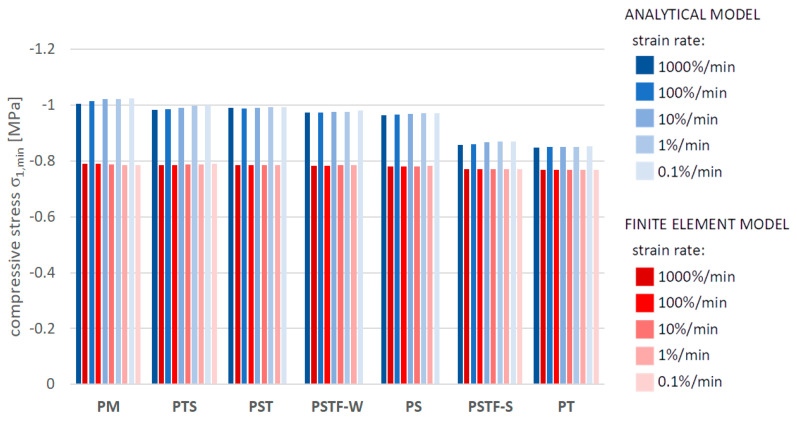
Minimal compressive stress in the mid-span in the top fibers of the top plate.

**Figure 15 polymers-16-00444-f015:**
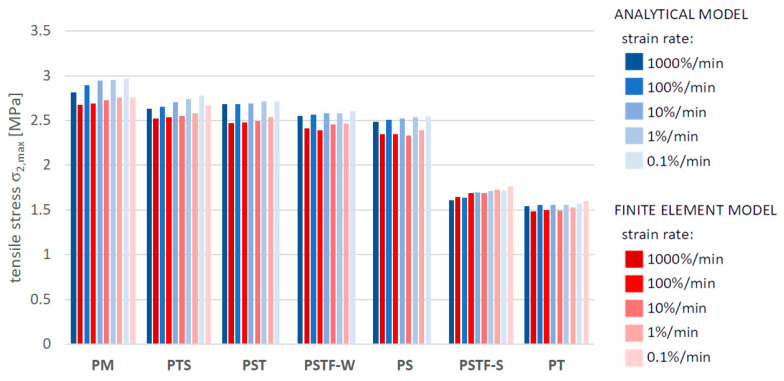
Maximal tensile stress in the mid-span in the bottom fibers of the bottom beam.

**Figure 16 polymers-16-00444-f016:**
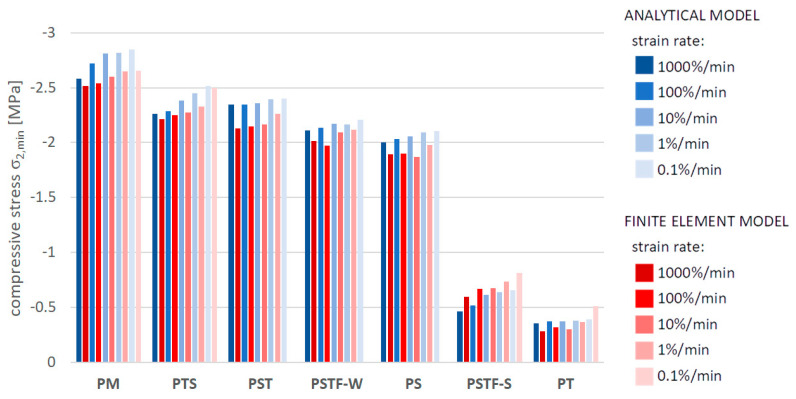
Minimal compressive stress in the mid-span in the top fibers of the bottom beam.

**Table 1 polymers-16-00444-t001:** Initial tangent Young’s moduli in MPa obtained from uniaxial tension test.

$\dot{\varepsilon }$	PM	PTS	PST	PSTF-W	PS	PSTF-S	PT
1000%/min	10.326	18.864	16.286	23.759	27.97	505.44	1128.9
100%/min	7.252	18.021	16.346	22.951	26.719	402.98	952.18
10%/min	5.5109	15.347	15.958	21.707	25.774	282.19	927.52
1%/min	5.3612	13.493	15.044	21.909	24.53	263.3	888.87
0.1%/min	4.7335	11.822	14.877	20.425	24.101	252.74	779.74

**Table 2 polymers-16-00444-t002:** Relative increment of initial tangent Young’s moduli with increasing strain rates [%].

	PM	PTS	PST	PSTF-W	PS	PSTF-S	PT
$\frac{E_{1000\%}-E_{0.1\%}}{E_{0.1\%}}\times 100$	118	60	9	16	16	100	45

**Table 3 polymers-16-00444-t003:** Magnitudes of the UDL for which first cracking occurs in the beam.

Model	PM	PTS	PST	PSTF-W	PS	PSTF-S	PT
Analytical	5.0	6.4	6.2	7.0	7.4	16.5	17.9
FEM	6.0	7.0	7.5	8.0	8.5	15.8	18.5

**Table 4 polymers-16-00444-t004:** Results of simulation of bending of a composite girder loaded with 20 kN/m^2^ UDL, taking into account cracking of concrete.

Model	w_max_ [mm]	τ_max_ [kPa]	σ_1,max_ [MPa]	σ_1,min_ [MPa]	σ_2,min_ [MPa]
Analytical	1.00	501	−0.342	−2.017	−0.0129
FEM	0.88	370 (260) *	−0.507	−1.547	−0.639

* In the FEM model, the maximal shear stress in the adhesive layer was found not at the end of the beam, which has a 20 cm overhang from the axis of support (the corresponding value is given in the brackets), but in the support zone.

## Data Availability

The data presented in this study are available on request from the corresponding author.

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
