# Peer review of "The Influence of the Flexibility of a Polymeric Adhesive Layer on the Mechanical Response of a Composite Reinforced Concrete Slab and a Reinforced Concrete Beam Girder"

_polymers, 2024, doi:10.3390/polym16030444_

Round 1

Reviewer 1 Report

Comments and Suggestions for Authors

Comments to authors are listed below: 

·         The abstract is very short and lacks to present the study's objectives and motivations and novelty  were not clearly stated.

·         The findings given in the abstract and conclusion are expectable facts. You can give quantitative results that may be comparable with other product.

·         The authors have not conveyed the strengths of the study and have not been supported by previous literature.

·         The results were depended on the mechanical properties but authors should support the strength of this work by carried out some mechanical characterisations such as the storage modulus (DMA) and flexural tests, and impact tests.

·         Figure 19-15 are not clear and should optimise them.

Comments on the Quality of English Language

English writing should be checked and improved. 

Author Response

Dear Reviewers, we would like to thank for your effort and time spent on reviewing the manuscript entitled “Influence of flexibility of a polymeric adhesive layer on mechanical response of a composite RC slab – RC beam girder”. Your remarks help us to improve the quality of the paper. We do hope that you will find our response satisfactory.

Reviewer #1

  1. The abstract is very short and lacks to present the study's objectives and motivations and novelty were not clearly stated.

Ad 1) The abstract was reformulated – specific objectives of the research were pointed out. The context of non-standardized design approach was also highlighted, which determines the novelty of this work (introduction - yellow highlighted in Abstract on Page 1).

  1. The findings given in the abstract and conclusion are expectable facts. You can give quantitative results that may be comparable with other product.

Ad 2) According to our best  knowledge no similar research was conducted as regards both polyurethanes as well as other polymers, since the analytical model investigated in the article is author’s own proposition, which is not widely recognized yet. First attempts in bonding of concrete-to-concrete elements of composite structures can be found in [25, 26]. Advantages and disadvantages of epoxy adhesives in concrete-to-concrete bonds were presented in [27], but any flexible polyurethane solution was not mentioned there, thus novelty of the proposed solution is valid. It was added in the Introduction chapter (introduction - yellow highlighted on Page 2).

Added references:[25] Yousef, R.F.; Al-Rubaye, M.M.; Muteb, H.H. Mechanical and chemical bond for composite action of precast beams. Curved Layer. Struct. 2022, 9, 304–319, doi:10.1515/cls-2022-0024.

[26] Alwash, N.A.; Abd Al-Radha, D.A. Theoretical Behavior of Composite Construction Precast Reactive Powder RC Girder and Ordinary RC Deck Slab. Int. J. Civ. Eng. Technol. 2015, 6, 08–21.

[27] Daneshvar, D.; Behnood, A.; Robisson, A. Interfacial bond in concrete-to-concrete composites: A review. Constr. Build. Mater. 2022, 359, 129195, doi:10.1016/j.conbuildmat.2022.129195

  1. The authors have not conveyed the strengths of the study and have not been supported by previous literature.

Ad 3) The benefits from the presented investigation are now highlighted in an additional paragraph in Conclusions section (introduction - yellow highlighted on Page 16). Employing the FPUJ as a load carrying element in composite newly designed bridge girders is a relatively new field of research. The analytical model considered in the paper was previously investigated in [29, 33]. Application of FPUs in a number of engineering problems was discussed in e.g.:

  1. Kwiecień, A. Polymer Flexible Joints in Masonry and Concrete Structures, Monography No. 414, Series of Civil Engineering (in Polish); Wydawnictwo Politechniki Krakowskiej: Kraków, 2013;
  2. Kwiecień, K.; Kwiecień, A.; Stryszewska, T.; Szumera, M.; Dudek, M. Durability of PS-Polyurethane Dedicated for Composite Strengthening Applications in Masonry and Concrete Structures. Polymers (Basel). 2020, 12, 1–16, doi:doi:10.3390/polym12122830.
  3. Akyıldız, A.T.; Kowalska-Koczwara, A.; Kwiecień, A. Stress distribution in masonry infills connected with stiff and flexible interface. J. Meas. Eng. 2019, 7, 40–46, doi:10.21595/jme.2019.20449.
  4. De Santis, S.; Stryszewska, T.; Bandini, S.; de Felice, G.; Hojdys, Ł.; Krajewski, P.; Kwiecień, A.; Roscini, F.; Zając, B. Durability of steel reinforced polyurethane-to-substrate bond. Compos. Part B Eng. 2018, 153, 194–204, doi:10.1016/j.compositesb.2018.07.043.
  5. Kozak, A.; Kwiecień, A. Accelerated weathering tests of polyurethane mass for flexible joints to repair concrete and masonry structural elements. Proceedings of the 68th Krynicka Scientific Conference of the Committee of Civil and Water Engineering of the Polish Academy of Sciences and the Science Committee PZITB, Gliwice, Poland, 24-28 September 2023; 2023.
  6. Kwiecień, A. Shear bond of composites-to-brick applied with highly deformable, in relation to resin epoxy, interface materials. Mater. Struct. Constr. 2014, 47, 2005–2020, doi:10.1617/s11527-014-0363-y.
  7. Cruz, J.R.; Sena-Cruz, J.; Rezazadeh, M.; Seręga, S.; Pereira, E.; Kwiecień, A.; Zając, B. Bond behaviour of NSM CFRP laminate strip systems in concrete using stiff and flexible adhesives. Compos. Struct. 2020, 245, 1–18, doi:10.1016/j.compstruct.2020.112369.
  8. Kwiecień, A.; Krajewski, P.; Hojdys, Ł.; Tekieli, M.; Słoński, M. Flexible adhesive in composite-to-brick strengthening-experimental and numerical study. Polymers (Basel). 2018, 10, 1–23, doi:10.3390/polym10040356.
  9. Kwiecień, A. Stiff and flexible adhesives bonding CFRP to masonry substrates-Investigated in pull-off test and Single-Lap test. Arch. Civ. Mech. Eng. 2012, 12, 228–239, doi:10.1016/j.acme.2012.03.015.

Since most of the research devoted to FPUs in composite civil engineering structures is conducted by Authors and their teams, adding these articles in the bibliography makes the value of self-citation to total citation ratio unacceptable by the MDPI editors (in fact, some positions in the bibliography had to be scrapped from the initial submission).

  1. The results were depended on the mechanical properties but authors should support the strength of this work by carried out some mechanical characterisations such as the storage modulus (DMA) and flexural tests, and impact tests.

Ad 4) The loads applied to bridge structures are predominantly quasi-static loads. In case of non-slender structures the dynamic load component is often surpassed by structure’s self-weight and live load of quasi-static nature. For these reasons neither storage modulus nor dynamic (impact) stiffness would be appropriate for predicting the mechanical response of such a structure. Proper remark on this issue was added in the manuscript. Uniaxial tension test may be also employed in the FEA software in a straightforward manner in order to specify the parameters of the constitutive model. Anyway, DMA tests on this type of polyurethanes were carried out in cases of polymer bearings tested as a seismic isolation system [31]. More details about other material properties of the considered polyurethanes can be found in [32] (introduction - yellow highlighted on Pages 3-4).

Added references:

[31] Falborski, T.; Jankowski, R. Experimental study on effectiveness of a prototype seismic isolation system made of polymeric bearings. Appl. Sci. 2017, 7, doi:10.3390/app7080808

[32] Śliwa-Wieczorek, K.; Zając, B. PUFJ (PolyUrethane Flexible Joints) as an innovative polyurethane system for structural and non-structural bonding of timber elements. J. Phys. Conf. Ser. 2023, 2423, doi:10.1088/1742-6596/2423/1/012015.

  1. Figure 19-15 are not clear and should optimise them.

Ad 5) The legend in all graphs was improved (changes - on Pages 11-14).

  1. English writing should be checked and improved.

Ad 6) English language was improved in the text.

Reviewer 2 Report

Comments and Suggestions for Authors

The reviewed work shows that a layer of polyurethane adhesive can be used to connect a reinforced concrete slab to a reinforced concrete T-profile. Such a flexible connection allows for an intermediate situation between a rigid connection and free sliding. To describe this connection, a simplified mathematical model has been developed, which agrees with the results obtained by the finite element method and uniaxial tensile tests with acceptable accuracy (5-20%). In general, this work has been completed sufficiently and can be of practical interest. Therefore, I can recommend it for publication.

Author Response

Dear Reviewers, we would like to thank for your effort and time spent on reviewing the manuscript entitled “Influence of flexibility of a polymeric adhesive layer on mechanical response of a composite RC slab – RC beam girder”. Your remarks help us to improve the quality of the paper. We do hope that you will find our response satisfactory.

Reviewer 3 Report

Comments and Suggestions for Authors

Paweł Szeptyński et al., in their manuscript entitled "Influence of flexibility of a polymeric adhesive layer on mechanical response of a composite RC slab – RC beam girder", prepared and studied a composite girder, which consists of a reinforced concrete (RC) slab and RC beam connected with a layer of flexible polyurethane (FPU) adhesive. They investigated the influence of the mechanical properties of adhesive on the girder’s deflection and on maximum stresses in both adhesive and concrete. However, some of the serious limitations listed below were observed:

1.      The manuscript contains many typos and spelling mistakes. They should be corrected.

2.      The abstract is very general. The authors should reformulate it according to their results and work novelty.

3.      In the introduction, the authors wrote, “As an innovative connection between reinforced concrete (RC) slab and RC T-section, the use of a layer of flexible polyurethane adhesive (FPU) is proposed [15].” What is the novelty of their work if they have already published their work?

4.      The last paragraph in the introduction should be rearranged in the conclusion.

5.   The authors should add information about the used polyurethanes (grade, characteristics, etc.).

6.      The scientific discussion of the results in section “2.3. Results of the uniaxial tension test” is absent. The authors should explain the difference in the obtained mechanical properties.

7.   The authors wrote “The analytical model underestimates the maximal deflection. The relative difference is 5-20%. The greatest discrepancies may be observed for stiff polyurethanes.” The relative difference of 20% is considered high; can the authors explain their results?

8.      Why can the proposed model be important for readers? 

Comments on the Quality of English Language

Moderate editing of the English language is required.

Author Response

Dear Reviewers, we would like to thank for your effort and time spent on reviewing the manuscript entitled “Influence of flexibility of a polymeric adhesive layer on mechanical response of a composite RC slab – RC beam girder”. Your remarks help us to improve the quality of the paper. We do hope that you will find our response satisfactory.

Reviewer #3

Paweł Szeptyński et al., in their manuscript entitled "Influence of flexibility of a polymeric adhesive layer on mechanical response of a composite RC slab – RC beam girder", prepared and studied a composite girder, which consists of a reinforced concrete (RC) slab and RC beam connected with a layer of flexible polyurethane (FPU) adhesive. They investigated the influence of the mechanical properties of adhesive on the girder’s deflection and on maximum stresses in both adhesive and concrete. However, some of the serious limitations listed below were observed:

  1. The manuscript contains many typos and spelling mistakes. They should be corrected.

Ad 1) English language was improved in the text.

  1. The abstract is very general. The authors should reformulate it according to their results and work novelty.

Ad 2) The abstract was reformulated – specific objectives of the research were pointed out (introduction - yellow highlighted in Abstract on Page 1). The context of non-standardized design approach was also highlighted, which determines the novelty of this work (introduction - yellow highlighted on Page 2).

  1. In the introduction, the authors wrote, “As an innovative connection between reinforced concrete (RC) slab and RC T-section, the use of a layer of flexible polyurethane adhesive (FPU) is proposed [15].” What is the novelty of their work if they have already published their work?

Ad 3) Please note, that [15] is a conference paper published just few months ago. The innovation of the proposed solution follows from the fact, that flexible adhesive connections in composite girders are still within the scope of academic research rather than engineering practice (introduction - yellow highlighted on Page 2), since there are no standardized design rules for such connections. In [15] only theoretical model was presented and the experimental results were processed with the use of different assumptions. In the present article, both theoretical and numerical (FEM) models are considered and compared. Another novelty is that cracking of the concrete was also involved in the analysis. Proper comment was added in the text (introduction - yellow highlighted on Pages 2-3).

  1. The last paragraph in the introduction should be rearranged in the conclusion.

Ad 4) The last paragraph in the Introduction contains mostly the description of the content of the article. Some general remarks concerning the FPU were also added in the Conclusions (introduction - yellow highlighted on Page 16).

  1. The authors should add information about the used polyurethanes (grade, characteristics, etc.).

Ad 5) Literature on properties of considered polyurethanes was provided in section 2.1 (introduction - yellow highlighted on Page 3).

  1. The scientific discussion of the results in section “2.3. Results of the uniaxial tension test” is absent. The authors should explain the difference in the obtained mechanical properties.

Ad 6) Comments on the obtained results was added in section 2.3 (introduction - yellow highlighted on Page 5).

  1. The authors wrote “The analytical model underestimates the maximal deflection. The relative difference is 5-20%. The greatest discrepancies may be observed for stiff polyurethanes.” The relative difference of 20% is considered high; can the authors explain their results?

Ad 7) There are three main sources of these discrepancies:

  • Analytical model employs the Bernoulli’s hypothesis of plane cross-sections, while the 3D FEM model is not restricted by such an assumption
  • Analytical model assumed that the adhesive layer undergoes simple shear only – it disregards transverse contraction and longitudinal elongation of the adhesive. These phenomena are accounted for in the FEM model.
  • In the analytical model the constitutive relations for the adhesive are assumed to be linear (Hooke’s Law), while in the numerical model the nonlinear hyperelastic Mooney-Rivlin constitutive model is introduced.

Appropriate comment was added in the manuscript (introduction - yellow highlighted on Page 15).

  1. Why can the proposed model be important for readers?

Ad 8) The proposed analytical model provides closed-form expressions for most important quantities taken into consideration during the design process, namely: maximal deflection, maximal stresses etc. Since the discrepancies between the simplified and detailed model are not very large, these design formulas may be used as the first approximation of the final dimensions of a composite structure. While final dimensions of the structure must be determined with the use of an appropriately precise model (in most of cases a FEM numerical model), this very first approximation requires nothing more than just substituting the model parameters in the formulas (introduction - yellow highlighted on Page 16).

Round 2

Reviewer 1 Report

Comments and Suggestions for Authors

Comments to authors are listed below:

1.      The abstract still is required to be strengthened with significant numerical values.

2.      The applications and novelty of this work in the last paragraph still did not state clearly.

3.      The discussion of results lacks to present the data in detail and compare with previous literature to show the differences and improvements.

Comments on the Quality of English Language

The English language of this paper should be checked and improved. 

Reviewer 3 Report

Comments and Suggestions for Authors

The author made sufficient improvements to the manuscript.